# What Is New in Morphea—Narrative Review on Molecular Aspects and New Targeted Therapies

**DOI:** 10.3390/jcm13237134

**Published:** 2024-11-25

**Authors:** Tomasz Stein, Paulina Cieplewicz-Guźla, Katarzyna Iżykowska, Monika Pieniawska, Ryszard Żaba, Aleksandra Dańczak-Pazdrowska, Adriana Polańska

**Affiliations:** 1Department of Dermatology, Poznan University of Medical Sciences, 60-806 Poznan, Poland; paulina.cieplewicz@gmail.com (P.C.-G.); adriana-polanska@usk.poznan.pl (A.P.); 2Institute of Human Genetics, Polish Academy of Sciences, 60-479 Poznan, Poland; katarzyna.izykowska@igcz.poznan.pl (K.I.); monika.pieniawska@igcz.poznan.pl (M.P.)

**Keywords:** morphea, localized scleroderma, pathogenesis, treatment

## Abstract

Morphea, also known as localized scleroderma, is an autoimmune chronic connective tissue disease. It is characterized by excessive collagen deposition in the dermis and/or subcutaneous tissue. The etiopathogenesis of this disease is not fully understood, with endothelial cell damage, immunological disorders, extracellular matrix disorders and factors such as infection, trauma and other autoimmune diseases being considered. As medicine advances, there is increasing evidence that genetic factors play a significant role in disease risk and progression. In addition to environmental factors and genetic predisposition, epigenetic factors may be potential triggers for morphea. Epigenetics studies changes that affect gene expression without altering the DNA sequence, such as microRNAs, long non-coding RNAs or DNA methylation. Understanding the pathogenesis of this disease is key to identifying potential new treatments. There are anecdotal reports of good therapeutic effects following the use of biological drugs such as tocilizumab, a humanized IgG monoclonal antibody; abatacept, a recombinant soluble fusion protein; JAK inhibitors, such as tofacitinib and baricitinib; and a drug used successfully in cancer treatment, imatinib, a tyrosine kinase receptor inhibitor. In this article, we aim to review up-to-date knowledge on the pathogenesis of morphea, with particular emphasis on genetic and epigenetic factors. In addition, we present the new options of morphea treatment based on several case series treated with new drugs that are potential targets for the development of therapies for this disease.

## 1. Introduction

Morphea, also known as localized scleroderma, is a chronic autoimmune connective tissue disease with an incompletely understood etiopathogenesis. It is characterized by excessive collagen deposition in the dermis and/or subcutaneous tissue [1]. The disease may also affect deeper tissues such as adipose tissue, fascia, muscle and bone [2]. The clinical picture usually shows an erythematous inflammatory patch and waxy yellow foci of induration [3]. The exact cause that initiates the disease process is still unknown. Certain stimuli such as drugs/infection or trauma lead to dysregulation of angiogenesis and the immune system in genetically predisposed individuals [4]. In particular, T lymphocyte activation and the release of interferon-γ-related cytokines are involved, resulting in the activation of inflammatory and profibrotic pathways leading to excessive collagen production [5,6].

Morphea is a rare autoimmune connective tissue disease, and its incidence appears to be underestimated. The only population-based study conducted by Peterson et al. in Olmsted County, Minnesota, USA, between 1960 and 1993 suggests that the incidence of morphea is approximately 0.4–2.7 cases/100,000 [7]. The disease is 2.6–6 times more common in women than in men [8]. Based on retrospective studies, morphea appears to be more common in Caucasians [9,10]. The incidence is similar in children and adults. There are two peaks of morphea incidence, the first between 2 and 14 years of age and the second in adults between 40 and 50 years of age [11].

Several forms of morphea have been described, each with different clinical manifestations and degrees of connective tissue involvement. In 2009, Kreuter et al. proposed a division into five main clinical types of morphea: limited, generalized, linear (which includes the linear limb variant, en coup de sabre and hemifacial atrophy), deep and mixed [12]. This classification is supported by the European Dermatological Forum [13] and the Polish Dermatological Society [14]. In adults, the most common clinical form of morphea is the plaque-like variety (morphea en plaques, MEP) [15], while in children it is the linear variety [16].

Systemic sclerosis, which is a distinct entity, should be considered in the broad differential diagnosis. Features suggestive of systemic sclerosis include a mask-like face, microstomia, telangiectasias, Raynaud’s sign, sclerodactyly, finger ulcers, the presence of antibodies (anti-Scl-70 and anti-centromere) and the involvement of internal organs [3,17].

Very similar clinically to morphea is lichen sclerosus. Morphea may coexist with lichen sclerosus (LS), especially with its extragenital form. Coexistence of lichen sclerosus and morphea in one lesion has been described clinically, histopathologically and dermoscopically [18]. Kreuteur et al. observed the coexistence of lichen sclerosus and morphea on histopathologic examination in 5.7% of morphea patients [19].

LS similar to morphea is a chronic, inflammatory dermatosis with an incompletely explained pathogenesis with a predilection for the anogenital area. Genetic, autoimmune, environmental, infectious and hormonal factors are believed to play a role. The presence of inflammation caused by chronic irritation, trauma and moisture in the genital area is also important [20]. The incidence is underestimated; it is assumed that in both sexes it may be 0.1% to 0.3% [21]. LS can occur at any age and in both sexes, with the ratio of women to men being as high as 10:1 [22]. In women, there are two peaks of LS incidence—the first before puberty, the second in the peri- and postmenopausal period, which is associated with low estrogen levels in these periods. However, it is currently believed that LS can occur in women of any age [23,24]. In contrast, in men, two peaks of occurrence are observed—in the third and sixth decades of life [25].

LS occurs in genital and extragenital forms, with 85% of patients having lesions in the genital mucosa [26].

In women, lesions are most often located in the prepuce of the clitoris, labia minora and majora, perineum and perianal area. These are hyperkeratotic, waxy, ivory-colored lumps that merge with each other and are accompanied by intense itching [20]. The most important complication of untreated lesions is scarring, leading to narrowing of the vaginal opening, pain and dyspareunia [24]. In prepubertal girls, the clinical picture is similar to that in adult patients, but erosions, petechiae and fissures are more frequently observed [24].

LS in males is usually located on the foreskin and glans penis, and anus involvement is rare. The occurrence of porcelain-white papules leads to phimosis, which is observed in almost 100% of boys with LS. Additionally, scarring can lead to narrowing of the external urethral opening and, as a consequence, dysuria and sexual dysfunction [27].

Skin lesions in the extragenital area are rarely observed, occurring mainly in women. They are usually asymptomatic, ivory-colored lumps located in the submammary region, neck, arms, inner thighs, wrists, and upper back in places of pressure or trauma [26].

LS is diagnosed based on the typical clinical picture; a biopsy of skin lesions for histopathological examination is performed in doubtful situations or when malignant changes are suspected [22].

The differential diagnosis of morphea is presented in Table 1.

Imaging methods can be used to monitor morphea. Ultrasonography is useful for assessing the extent and depth of the disease process and enables measurement of the thickness of the dermis and adipose tissue [28]. High-frequency ultrasonography (HF-USG) can be used to monitor the dynamics of the disease process. In the sclerosis phase, an increase in echogenicity and thickening of the skin is observed. In the inflammatory phase, an echogenicity decrease is observed [28,29]. Salgueiro et al. [30] observed an ultrasound symptom called the “sun sign”. This is a hyperechoic halo surrounding the superficial subcutaneous veins of the limbs in cross-section. There are interesting anecdotal data on the use of several other noninvasive methods like infrared thermography (IRT, skin temperature), laser Doppler imaging (LDI, skin blood flow) and multispectral imaging (MSI, oxygenation) in morphea, although their use requires further research [31].

Depending on the clinical form of morphea, its activity or severity, the depth and extent of the lesions and the degree of tissue damage, local and systemic treatments are used in therapy [4]. Decisions must take into account the chronic and recurrent nature of the disease and the presence of atrophy and extracutaneous complications, which can lead to significant disability [14] (Figure 1).

## 2. Pathogenesis

Morphea is a chronic disease whose pathogenesis is not fully understood. Several factors may play a role in the development of the disease, including genetic, epigenetic or environmental factors [2]. Damage to the endothelial cells is thought to be of primary importance in the pathogenesis of the disease and may initiate the disease process [32]. The following insults are mentioned: trauma, infection, radiation or drugs (β-blockers, bleomycin, bromocriptine, D-penicillamine). They cause microvascular damage and secondary activation of T lymphocytes [13]. These, in turn, release cytokines and growth factors that play a key role in the fibrotic process. Studies have confirmed that transforming growth factor β (TGF-β), interleukin IL-1β, IL-4, IL-6, IL-10, IL-27, interferon gamma (INF-γ) and, more recently, IL-17A are essential for inflammation and fibrosis [33,34,35,36,37,38]. Ultimately, activation of these pro-inflammatory and fibrosis-inducing mediators leads to excessive collagen production and reduced production of metalloproteinases responsible for collagen degradation [13].

In Figure 2, we show the pathogenesis of morphea.

### 2.1. Endothelial Cell Damage

The most likely earliest phenomenon underlying the pathogenesis of morphea is thought to be endothelial cell activation resulting from endothelial damage [39]. Endothelial cell activation, together with an increase in endothelin-1 production and a decrease in prostacyclin production, leads to vasoconstriction. Other factors contributing to vasculopathy include an imbalance between pro- and anti-angiogenic mediators, the presence of fibroblast-derived anti-angiogenic factors, and reduced levels of erythropoiesis-associated factor (Friend leukemia integration factor 1, Fli1), which is responsible for regulating immune function and collagen production. Damaged endothelial cells release cytokines that increase the production of adhesion molecules such as intracellular adhesion molecule (ICAM)-1, vascular cell adhesion molecule (VCAM)-1, E-selectin, and P-selectin. Increased levels of adhesion molecules, in turn, lead to increased recruitment of T lymphocytes, which then release cytokines that stimulate the fibrotic process, including IL-4, IL-6, and TGF-β [4]. Damage to the endothelial cells leads to obliteration of the lumen of small blood vessels and release of cellular antigens, which may be the initial trigger for inflammatory and autoimmune responses. The vascular theory in the pathogenesis of morphea is supported by the histopathological picture of the lesional skin. In the inflammatory phase of morphea, endothelial cells are swollen and activated, and there is thickening of the vessel walls surrounded by an inflammatory infiltrate and active fibroblasts. In the late phase of morphea, basement membrane proliferation and a decrease in vascular density are observed [40].

### 2.2. Immune Dysfunction

An important component in the pathogenesis of morphea is the activation of the immune system. Increased production of adhesion molecules such as ICAM-1, VCAM-1, E-selectin and P-selectin leads to increased recruitment of inflammatory cells, including T lymphocytes, monocytes and other immune cells [41]. Studies using flow cytometry in morphea patients (adults and children) have shown that the immune response is dominated by CD4+ helper T cells (Th) and there is a decrease in regulatory T cells (Treg) [42,43,44]. Histopathological examination of skin lesions taken during the early inflammatory phase of morphea has also shown activation of predominantly T lymphocytes, which may indicate their involvement in tissue damage and fibrosis induction [40]. Studies suggest the existence of an imbalance in the Th lymphocyte population, with a predominance of Th2 over Th1 lymphocytes, which also favors fibrotic processes [44]. Ihn et al., analyzing sera from patients with morphea and SSc, showed elevated levels of CD30 (Th2 lymphocyte surface antigen), which may also indicate that the immune response is associated with increased Th2 lymphocyte activity [5]. It has been postulated that the early, active phase of morphea is dominated by Th1 and Th17 lymphocyte responses. Th1 lymphocytes produce IFN-γ and IL-2, which are stimulated by IL-12 [45]. Cytokines such as interferon gamma-induced protein 10 (IP-10), monocyte chemotactic protein-1 (MCP-1), IL-17a, IL-12p70, granulocyte-macrophage colony-stimulating factor (GM-CSF), platelet-derived growth factor-beta, platelet-derived growth factor-bb (PDGF-bb), IFN-α2 and IFN-γ, measured in peripheral blood mononuclear cells (PBMCs), were increased in morphea patients compared to controls [46]. IL-12 stimulates Th1 lymphocytes to produce IFN-γ, which induces the expression of IP-10 [46]. IP-10 is produced by fibroblasts and endothelial cells, among others. IP-10 is thought to be a chemokine associated with the stimulation of a pro-inflammatory, Th1-dependent response and is positively correlated with disease activity. It has been suggested that this protein may be a future serological marker of disease activity [46]. Th17 lymphocytes produce IL-17A, IL-21, IL-22 [44]. Dańczak-Pazdrowska et al. demonstrated increased IL-17A gene expression in PBMCs from morphea patients compared to controls and a negative correlation between disease duration and IL-17A gene expression [34]. O’Brien et al. also demonstrated increased IL-17A gene expression in PBMC and skin of morphea patients and showed a positive correlation between increased IL-17A gene expression and shorter disease duration [47]. Torok et al. conducted a study in a group of 69 children diagnosed with morphea and showed significantly higher plasma levels of IP-10 and IL-17A and other cytokines—MCP-1, IL-12p70, GM-CSF, PDGF-bb, IFN-α2 and IFN-γ—in morphea patients compared to controls. In summary, the results of this study suggest that a pro-inflammatory response dependent on Th1 (IFN-γ) and Th17 (IL-17A) lymphocytes is more predominant in morphea than in SSc. Increased levels of these cytokines correlated with shorter disease duration. In addition, IL-4, IL-5 and IL-13 secreted by Th2 lymphocytes were not significantly increased [46]. It has been postulated that a Th2 lymphocyte-dependent inflammatory response predominates in the sclerosing phase of morphea [41]. Th2 lymphocytes produce IL-4, IL-5, IL-6, IL-10 and IL-13, which correlate with fibrotic processes [46]. IL-4 induces the production of TGF-β, which not only stimulates collagen production but also induces the differentiation of fibroblasts into smooth muscle cell myofibroblasts. The accumulation of myofibroblasts enhances the fibrotic process by increasing the stiffness of the extracellular matrix [48]. In addition, TGF-β stimulates the production of type I collagen, type III collagen and other extracellular matrix proteins by fibroblasts [49]. In addition, it has been suggested that the distribution of CD34+ skin dendritic cells (DCs) is altered during the sclerosing phase. CD34 expression in the stroma was significantly lower in patients with morphea than in healthy controls [50]. Some studies have reported significant numbers of plasmocytoid DCs in skin lesions in the deeper layers of the skin, around blood vessels and around collagen fibers in the subcutaneous tissue. They are thought to play an important role in both the development of immune tolerance mechanisms and the activation of autoreactive T cells [51].

### 2.3. Disorders of the Extracellular Matrix

The main regulator of the fibrosis process is TGF-β [40]. An increase in its concentration causes an increase in the expression of collagen types I, III, VI, X, fibronectin and proteoglycans [6,52]. It also causes a decrease in the production of proteases and an increase in their inhibitors [6]. Extracellular matrix metalloproteinases (MMPs) are also inactivated. Tomimura et al. showed that anti-MMP antibodies were present in patients with morphea and SSc, but not in healthy controls [53]. The role of insulin-like growth factor (IGF) in the pathogenesis of morphea has also been highlighted [4]. Fawzi et al. showed increased IGF gene expression in the skin of morphea patients compared to controls [54]. IGF increases collagen production, recruits fibroblasts and activates excessive extracellular matrix production [54].

### 2.4. Genetics Factors

Understanding the genetic basis of morphea is critical to uncovering the mechanisms underlying this disease and identifying potential treatment targets. Although most cases of morphea are sporadic, reports of familial aggregation suggest a potential hereditary component [55]. This means that genetic factors may contribute to susceptibility and play a significant role in the development and progression of morphea.

#### 2.4.1. HLA Involvement 

Human leukocyte antigen (HLA) has been implicated as the most promising target in the pathogenesis of morphea, suggesting a genetic predisposition to the disease [2]. HLA molecules are cell surface glycoproteins that primarily function to present both endogenous and exogenous antigens to T lymphocytes, which facilitate their recognition and response, thereby regulating the immune system [56]. Several studies have identified specific HLA alleles that are more common in people with morphea compared to the healthy population. Papara et. al. summarized a comprehensive correlation of the most significant associations within HLA-DRB1*04:04 and HLA-B*37, emphasizing that the alleles associated with morphea were different from those associated with systemic sclerosis (SSc), indicating that morphea is immunogenetically distinct from SSc [2]. Moreover, the *HLA-class II—HLA-DQB1*02:01*, *HLA-DRB1*03:01* and *HLA-DQA1*03:00*, and *HLA-class I—HLA-C*08* and *HLA-C*15*—have also been implicated in morphea patients [57]. The presence of HLA alleles might affect the disease’s severity and progression, possibly via pathways involving autoimmunity and sustained inflammatory processes. Moreover, HLA alleles in morphea patients were described to be strongly associated with multiple sclerosis (MS) and autoimmune thyroid disease (AITD) [2]. As mentioned above, HLA-DRB1*04:04 has also been associated with an increased risk of RA in cohorts of patients with morphea, supporting the hypothesis that autoimmune aberrations are involved in the pathogenesis of the condition [57]. Another study investigating the gene expression profiles of juvenile morphea identified significant upregulation of several HLA class II genes, including HLA-DQA1, HLA-DQB1, and HLA-DRB1, which were highly correlated with both inflammation and fibrosis scores [58]. These HLA alleles were also identified in a case-control study comparing the peripheral blood of patients with morphea with a healthy control group, indicating susceptibility to morphea [58]. Additional HLA testing in juvenile SSc supports the notion that HLA-DRB1 and HLA-DQA1 are associated with greater susceptibility to scleroderma [58]. The overlap between morphea and SSc, together with the correlation of these HLA genes with inflammation and collagen deposition in the current study, highlights the interconnection of inflammation and fibrosis in morphea, further characterizing morphea as a fibrotic disease driven by inflammatory processes [58].

#### 2.4.2. Cutaneous Mosaicism

Recent studies have highlighted the involvement of cutaneous mosaicism as a possible factor in morphea’s pathogenesis [59]. Mosaicism refers to the presence of genetically distinct cell populations in an individual resulting from postzygotic mutations [60]. Two subtypes of mosaicism can be distinguished: genomic (caused by de novo mutation, altering the DNA sequence) or epigenomic (change in gene expression not correlated with DNA sequence alterations) [61]. This phenomenon suggests that somatic mutations occurring in embryogenesis may contribute to the development of morphea lesions. Studies have identified mosaic mutations in genes associated with pathways involved in skin development and immune system regulation [61,62,63,64], which potentially may have an impact on the localized scleroderma phenotype, and the presence of mosaicism highlights the possibly complex genetic landscape of morphea—mainly the linear subtype, which follows Blaschko lines during epidermal development, determining the linear distribution of fibrosis processes [65,66].

#### 2.4.3. Transcriptome Alternations

Morphea transcriptome analysis revealed a predominance of IFN-γ-driven T helper 1 immune dysregulation, with relatively few identified fibrotic pathways [67]. In particular, gene expression profiles in morphea-affected skin were found to be consistent with an inflammatory subset of systemic sclerosis (SSc), clearly distinct from the fibroproliferative subset of SSc [67]. Additionally, the not-affected skin of patients with diagnosed morphea did not have the pathological gene expression signatures observed in not-affected SSc skin samples. Further examination of the IFN-γ-mediated chemokines CXCL9 and CXCL10 revealed increased transcription in the skin but not in the blood serum [67]. Despite increased transcriptional activity in the skin, serum CXCL9 levels were elevated and associated with active, widespread skin involvement [67]. These findings suggest that morphea is primarily a skin-directed disorder characterized by dysregulation of the T helper 1 immune system, which contrasts with the fibrotic features and systemic transcriptional changes observed in SSc [67]. A different study, comparing pediatric morphea gene signatures with healthy controls, indicated a distinct inflammatory response gene signature (IRGS) expression pattern, emphasizing genes related to IFNα, IFNγ, and TNFα [68]. Gene set enrichment analysis (GSEA) revealed that this IRGS, which includes interferon-induced chemokines such as CXCL9, CXCL10, CXCL11, and IFNγ, was more pronounced in LS patients showing more inflammatory changes [68]. Moreover, elevated levels of CXCL9 and CXCL10 were detected in the serum of patients with morphea, consistent with increased transcription and expression of CXCL9 in skin lesions, and this increase has been linked to inflammation and disease activity [47,67].

### 2.5. Epigenetics Factors

Epigenetic factors are considered to be triggers of the morphea, next to environmental factors and genetic predisposition [59]. Epigenetics investigates changes that influence gene expression without altering the DNA sequence, like DNA methylation, post-translational modification of histones, microRNAs (miRNAs), and long non-coding RNAs (lnRNAs) [41].

#### 2.5.1. DNA Methylation

Abnormal DNA methylation resulting in altered gene expression has been shown in many autoimmune disorders, including systemic scleroderma, where both PBMCs and dermal fibroblasts were investigated [69]. Not many studies have been performed on morphea patients. The first and only genome-wide DNA methylation study in PBMCs isolated from a cohort of juvenile localized scleroderma (jLS) was performed by Coit et al. [70]. The methylome of jLS patients was compared not only to healthy individuals but also to juvenile systemic sclerosis patients (jSSc), and the majority of differentially methylated sites and genes were unique to either jSSc or jLS, suggesting differences in epigenetic patterns and inflammatory and cell signaling pathways in both diseases. In general, inflammatory pathways were enriched in genes deferentially methylated in jSSc, including STAT3, NF-kB, and IL-15 pathways, while the HIPPO signaling pathway, which regulates cell proliferation and contact activities, was enriched in jLS [70]. In both jSSc and jLS patients compared to controls, consistent hypermethylation in CpG sites within FGFR2 was identified. FGFR2 encodes the fibroblast growth factor receptor 2 protein that has broad biological functions including promoting cell proliferation, survival, migration, and adhesion. Overall, the analysis showed that PBMNc in jSSc and jLS are unique at the epigenetic level and that the epigenic pattern of jSSc appears to be more inflammatory compared to jLS [70].

#### 2.5.2. MicroRNA

MicroRNAs (miRNAs) are small non-coding RNAs that usually bind to complementary sequences in the 3′ untranslated regions (UTR) of mRNAs, which results in inhibiting their translation into protein. They are involved in the regulation of a variety of cellular processes like cell differentiation, proliferation, apoptosis, and immune response [71]. Although the studies on miRNAs in morphea are rather limited, many recent reports have shown that miRNAs are involved in the regulation of processes that are related to this disorder, like fibrosis, including alterations in the TGF-β signaling cascade (miRNA-18, miRNA-20, miRNA-21, miRNA-23b, miRNA-29, miRNA-140-5p, miRNA-146a, miRNA-206), fibroblast proliferation and differentiation (miRNA-21, miRNA-31, miRNA-146a, miRNA-200), and extracellular matrix synthesis and deposition (miRNA-let-7a, miRNA-7, miRNA-26a, miRNA-29, miRNA-129-5p, miRNA-133a, miRNA-133b, miRNA-150, miRNA-196a) [71,72].

Morphea is characterized by skin fibrosis due to excessive deposition of type I collagen, and studies have shown that several miRNAs are regulators of this collagen expression [73,74,75]. One of them, miR-7, was significantly downregulated in skin, dermal fibroblast, and serum of LSc patients compared with healthy controls [73]. Upon specific inhibition of miR-7 in cultured normal dermal fibroblasts, the upregulation of α2(I) collagen protein was observed. Similar studies were performed for miRNA-196a and let-7a [74,75], which are putative regulators of α1(I) and α2(I) components of type I collagen [71]. The analysis showed that the level of miRNA-196a was markedly decreased in LSc skin tissue and serum [75], and blocking the miR-196a with the specific inhibitor in normal cultures of human dermal fibroblasts led to the upregulation of type I collagen. Let-7a expression was also downregulated not only in the skin of LS patients but also in SSc; however, in terms of serum concentration, the level was significantly decreased especially in LSc patients [74]. The inhibition or overexpression of let-7a in human or mouse skin fibroblasts significantly affected the type I collagen protein expression. All those studies proved that downregulation of miR-7, let-7a, and miR-196a and subsequent overexpression of type 1 collagen in dermal fibroblasts may play a key role in the pathogenesis of LSc [73,74,75].

One of the upregulated miRNAs in patients with morphea and systemic scleroderma was miR-155 [76]. The expression was elevated in skin tissue but also in the experimental skin fibrosis mouse models. MiR-155−/− mice showed less fibrosis in the dermis when challenged with bleomycin in terms of skin thickness and collagen content [76]. Both local and systemic miR-155 silencing in vivo remarkably attenuated bleomycin-induced dermal fibrosis, while miR-155 silencing in primary skin fibroblasts inhibited collagen production. Molecular analysis in vitro showed that Wnt/β-catenin and Akt signaling, pathways necessary for fibrosis development, are regulated by miR-155 via direct targeting of CK1α (casein kinase 1α) and SHIP-1 (inositol phosphatase-1) [76]. This extensive analysis of miR-155 showed this miRNA as an important epigenetic player in scleroderma and a potential treatment target.

Another upregulated miRNA in conditions marked by skin fibrosis, including morphea, was the miR-483-5p [77]. Although the functional validation was performed on systemic sclerosis (SS), it proved that miR-483-5p modulates the expression of fibrosis-related genes involved in myofibroblast transition (αSMA, SM22A) and collagen production (Col4A1, Col4A2, Col1A2, Fli-1) in fibroblasts and endothelial cells. This study proposed that miR-483-5p serves as a new molecular marker for the early stages of the disease and contributes to the establishment of the fibrotic phase [77].

#### 2.5.3. Circulating MicroRNAs

Circulating microRNAs (c-miRNAs), which are present in almost all biological fluids, are promising and sensitive biomarkers for various diseases [78]. Their signatures could be used in the clinic to assess the patient state and treatment efficiency. In morphea patients, the serum levels of six miRNAs—miRNA-181b-5p, miRNA-223-3p, miRNA-21-5p, let 7i-5p, miRNA-29a-3p and miRNA-210-3p—were significantly increased compared to healthy individuals [79]. All of those miRNAs were either previously linked to morphea, SS, or other autoimmune disorders or were related to fibrosis [79]. The authors performed a series of correlation analyses between miRNA expression and the severity and clinical symptoms of morphea. The analysis revealed that, in the female morphea patients, some significant correlations between several miRNAs’ serum levels and the severity and other clinical symptoms of morphea were identified; for example, the increased level of let-7i co-existed with less severe morphea, while the miRNA-21 serum level was inversely correlated with inflammation [79]. The diagnostic and prognostic value of circulating miRNAs is worth further investigation.

#### 2.5.4. Long Non-Coding RNAs

Long non-coding RNAs (lncRNAs) are groups of RNA transcripts longer than 200 nucleotides that regulate gene expression at transcriptional and post-transcriptional levels [80]. The lncRNA profile of inflammatory cells was investigated in skin tissue samples from pediatric morphea and fibroblasts co-cultured with CD4+ T lymphocytes. The analysis revealed 11 candidate lncRNAs; among them, the MIR100HG that might be involved in the fibrosis, via the miR-29a-3p/Tab1/TGF-β1 axis, and ZEB1-AS1 via the ZEB1-AS1/miR-141-3p axis, respectively [80]. The involvement of the TGF-β receptor signaling pathway was also confirmed in the study by GO analysis based on the lncRNA expression profiling.

### 2.6. Other Factors in the Pathogenesis of Morphea

Infectious factors include the occurrence of morphea-like lesions following infections with measles, varicella, Epstein–Barr, hepatitis B and C viruses [81,82,83,84]. Other possible aetiological factors include infection with Borrelia burgdorferi (B. burgdorferi), the role of which remains controversial and debated. The association between morphea and Lyme disease was first suggested by Aberer et al. in 1987 [85]. Since then, numerous reports using both serological methods and PCR detection of B. burgdorferi DNA in a skin lesion have failed to confirm the association between morphea and B. burgdorferi infection [86,87,88,89]. Weide et al. report that late skin lesions in Lyme disease may be associated with pseudosclerotic changes, but these cannot be considered as changes in the course of morphea [90]. The association of morphea with other autoimmune diseases has been described in the medical literature for years. According to Leitenberger et al., 30% of adult patients with morphea have a coexisting autoimmune disease [10], a higher frequency than in the healthy population. The most common autoimmune diseases are plaque psoriasis, systemic lupus erythematosus, multiple sclerosis and vitiligo [8]. However, recent studies have not shown an increased frequency of positive antithyroid antibodies in adult patients with morphea [91], in contrast to the pediatric population where coexistence with autoimmune thyroiditis is significant [92]. Patients with the generalized form of morphea are much more likely to have co-existing autoimmune diseases than patients with other forms of morphea [8]. The medical literature also describes a more frequent co-occurrence of morphea with lichen sclerosus of the anogenital area, with some data suggesting a frequency of 38% [93]. Since 1990, cases of morphea-like changes have been described in patients following radiotherapy, mainly in patients with a history of breast cancer [94]. The pathogenesis of this phenomenon remains unclear. Data to date suggest that ionizing radiation activates fibroblasts to increase secretion of cytokines with profibrotic effects (IL-4, IL-5, TGF-β) [95]. There are also reports in the medical literature of cases of skin fibrosis associated with chemotherapy, following the use of drugs such as docetaxel, paclitaxel, bleomycin, peplomycin, fluoropyridines, gemcitabine, doxorubicin and cyclophosphamide [96] and associated with the use of checkpoint inhibitors [97]. Trauma has also been mentioned as a possible factor in the pathogenesis of morphea. Some authors postulate the occurrence of the Koebnerization phenomenon as an important factor in the pathogenesis of morphea [98,99]. The cause of this reaction is still unknown, but it is thought that cytokines, stress proteins, adhesion molecules and/or autoantigens are released into the skin under the influence of mechanical trauma [99]. There have also been reports of morphea as a reaction to tattooing, with skin lesions limited to the tattooed area [100].

## 3. Treatment

The treatment of morphea remains a challenge for clinicians because of the poor clinical response to treatment. The main therapeutic options, depending on the type of morphea, include topical steroids, topical calcineurin inhibitors, imiquimod, and in more advanced cases methotrexate, systemic steroids, mycophenolate mofetil, and phototherapy (PUVA, UVA1, UVB) [41].

Treatment decisions should consider the subtype of morphea, the disease’s activity, and the depth of the skin lesions. The earlier treatment starts, the less damage, such as limb deformity, will occur. Topical corticosteroids are the first-line treatment for patients with superficial peripheral lesions. Treatment is prolonged, sometimes requiring intermittent therapy or alternative therapy such as topical tacrolimus [2].

The first-line treatment for extensive or deeply infiltrating lesions is ultraviolet A (UVA1 phototherapy, 340–400 nm), which has greater tissue penetration than UVB [101].

For very active, widespread lesions, systemic treatment with corticosteroids (intravenous or oral) or methotrexate is required. The combination of systemic corticosteroids and methotrexate has the best effect in this group of patients and is the first-line treatment [41].

Mycophenolate mofetil may be an alternative to methotrexate but is rarely used in dermatological practice. Surgical treatment may be considered in patients with the en coup de sabre type [41].

In pediatric patients, similar to adults, treatment depends on the clinical form of morphea, activity, severity or extent of the disease process [102]. In the case of limited skin involvement, the treatment of choice is the external use of topical agents (e.g., calcineurin inhibitors or calcipotriol [103]). Local application of glucocorticosteroids is effective in the case of active lesions, but in the pediatric population their use should be limited due to the risk of atrophy [104]. Phototherapy—UVA-1 or narrowband UVB—is recommended as second-line treatment in children over 12 years of age [13]. When extensive tissue involvement occurs (linear, deep, generalized form), rapid implementation of general treatment is required. The first-line treatment is methotrexate in monotherapy or in combination with oral or intravenous glucocorticosteroids. In the absence of improvement or intolerance to the above-mentioned treatment, the use of mycophenolate mofetil is recommended [102].

In addition to the methods mentioned, we also have surgical procedures. According to European dermatological recommendations, surgical procedures are indicated mainly in the linear form of limbs to correct differences in limb length [13]. Surgical interventions can also be considered in the case of cosmetic defects in the form of en coup de sabre or Parry–Romberg syndrome. To reduce the risk of disease reactivation, surgery should only be performed in the inactive form of morphea. There are no recommendations specifying the appropriate duration of such treatment; it is recommended that the disease be inactive for 3–5 years [105]. Recent studies indicate that in the case of significant esthetic or functional defects in the course of morphea, autologous fat transplantation can be considered [2]. It has also been shown to have anti-inflammatory and antifibrotic effects by reducing the expression of TGF-b1 and collagen type III [106,107]. Due to individual case reports, further research on this treatment method is needed.

As our understanding of the pathogenesis of this disease has progressed and genetic and autoimmune processes have been identified, we will focus on new therapeutic options using biological drugs, antifibrinolytic and small molecule agents.

### 3.1. Biological Drugs and Small Molecules

IL-6 plays a crucial role in the pathogenesis of morphea. It stimulates collagen production by fibroblasts and the differentiation of CD4+ cells into pathogenic Th17 cells, making it a pro-inflammatory and profibrotic interleukin [108].

Kreuter et al. [109] showed that IL-6 levels are elevated in the skin and serum of patients with morphea.

Tocilizumab is a humanized monoclonal antibody approved by the FDA for treating rheumatoid arthritis, giant cell arteritis, polyarticular juvenile idiopathic arthritis, systemic juvenile idiopathic arthritis and COVID-19 pneumonia [110].

In a randomized trial of subcutaneous tocilizumab in patients with systemic scleroderma, one effect was a reduction in skin thickness compared with placebo [111].

Based on this, it was hypothesized that the drug might play a role in patients with refractory morphea. The drug has shown promising results in several case series.

Lythgoe et al. [112] used tocilizumab in five pediatric patients with linear morphea. These were patients with an active phase of the disease, despite escalation to maximum tolerated doses of methotrexate and one other disease-modifying drug or biological drug.

The dosing regimen was 8 mg/kg for children weighing 30 kg or more and 12 mg/kg for children weighing less than 30 kg, administered as 0-, 2- and 4-week courses followed by 4-week intervals. Patients received the drug intravenously and were treated for 12 to 25 months. All patients tolerated the drug without serious adverse effects. Two of five patients required steroids at the start of tocilizumab therapy, and one patient resumed methotrexate and mycophenolate mofetil. All patients showed improvement in Physician Global Assessment (PGA) disease activity scores at 6 months. However, changes in the modified Localized Skin Severity Index (mLoSSI) were not statistically significant. The skin damage that occurred before tocilizumab treatment is largely irreversible, so future studies should focus on the pediatric population in the early stages of the disease. The authors suggest that intravenous administration of tocilizumab is safe [112].

Zhang et al. [113] described the case of a 6-year-old female patient treated with tocilizumab. The patient had a severe form of morphea, pansclerotic scleroderma. Before tocilizumab, she was treated with methotrexate, prednisone and mycophenolate mofetil. Due to disease progression and progressive fibrosis, mycophenolate mofetil was discontinued and intravenous tocilizumab 300 mg every four weeks was started, while methotrexate was continued at 25 mg weekly. Within a few months, there was significant improvement. The mLoSSI improved from 22 to 6 and the PGA-A from 30 to 17. Pansclerotic scleroderma is an extremely rare subtype of morphea characterized by an aggressive course and generalized thickening of the skin that can lead to joint deformity and, consequently, a significant reduction in joint mobility. The combination of tocilizumab and methotrexate, and therefore the efficacy of this therapy, requires further study in a larger group of patients.

Martini et al. [114] described the cases of two pediatric patients with pansclerotic scleroderma who, after the failure of disease-modifying drugs, were treated with tocilizumab, which reduced disease activity and halted disease progression. These authors suggest that tocilizumab should be used in the early stages of the disease to prevent tissue damage. The descriptions above were related to pediatric patients.

Lonowski et al. [115] presented three adult women between 41 and 69 years; one woman had linear morphea (en coup de sabre) and two women had generalized morphea. The drug was administered intravenously or subcutaneously according to patient preference (subcutaneously at 162 mg/week or intravenously at 8 mg/kg/month). One patient with linear morphea was also treated with mycophenolate mofetil, and one patient with generalized scleroderma was treated with intravenous immunoglobulin. None of the patients required steroids at baseline or during tocilizumab therapy.

Improvement of scleroderma was observed in all patients. CR (Complete remission) was achieved in two patients and partial improvement in three patients. The drug was well tolerated.

Another potential target is the JAK/STAT pathway. In vitro and mouse studies have shown that JAK inhibitors can effectively reduce or even block TGF-β-induced skin fibrosis [116].

The JAK-STAT pathway is thought to mediate the promotion of fibrosis by TGF- β, IL-6 and other factors, as well as play a role in signaling in the fibrotic process [117].

Tang et al. [118] described the use of tofacitinib, a JAK1 and JAK3 inhibitor, in two pediatric patients with limited morphea. The 6-year-old girl was initially treated with glycyrrhizin, topical tacrolimus and UVA1. The patient’s parents did not agree to treatment with methotrexate and mycophenolate mofetil, so the patient was started on tofacitinib at an initial dose of 2.5 mg 2 × day for 4 months, followed by 2.5 mg every other day for a further 2 months. The drug was well tolerated. After 6 months, the mLoSSI improved from 6 to 1 and LosDi (Localized scleroderma damage index) from 7 to 2. In the second patient, a 13-year-old boy diagnosed with linear morphea, hydroxychloroquine, topical tacrolimus and UVA1 were used; his parents refused methotrexate and corticosteroids. Due to disease progression, tofacitinib was added to the treatment regimen—5 mg 2 × day for 2 months and then, due to the rapid clinical response, the dose was reduced to 5 mg per day for a further 3 months. After 5 months, the mLoSSI improved from 7 to 1 and the LosDi from 7 to 2. In both cases, the skin sclerosis resolved, which was confirmed by histopathology and immunofluorescence, where an upregulation of collagen synthesis was shown to be the basis of the treatment.

Baricitinib, a JAK1 and JAK2 inhibitor, has shown a positive effect in a patient with generalized morphea [119].

Another drug, abatacept, is a fusion protein consisting of the extracellular domain of human cytotoxic T-lymphocyte-associated antigen-4 (CLTA-4) fused to a modified Fc fragment of human immunoglobulin G1 (IgG1). It is produced by recombinant DNA technology in Chinese hamster ovary cells [120]. The drug selectively modulates a key costimulatory signal required for full activation of T cells expressing CD28. Abatacept has a more substantial effect on the response of T lymphocytes that are not in contact with any stimulus than on the response of memory T lymphocytes. In vitro studies show that the drug weakens T lymphocytes by reducing their proliferation rate and ability to produce cytokines [120]. It reduces the production of interferon-gamma, interleukin-2 and TNFα by T lymphocytes, and reduces the plasma concentration of interleukin-6, a product of activated macrophages; thus, the drug may reduce skin fibrosis [120].

Stausbøl-Grøn et al. [121] described two patients with deep morphea treated with abatacept, one of whom showed significantly improved skin and mobility. Both were able to reduce their steroid doses.

Fage et al. [122] presented 13 patients with a refractory, severe form of limited scleroderma. Patients were treated with abatacept with a mean time between disease severity assessments of 16 months, a relatively short interval. Improvement and reduction in lesion size and improvement in the mLoSSI were observed during treatment.

Li et al. [123] retrospectively evaluated the safety and efficacy of abatacept in patients with refractory limited morphea. Patients were followed for 12 to 24 months, with clinical assessments at 6-month intervals, including skin and PGA-A measurements. There was a cohort of 18 patients with a mean age of 13.4 years, most of whom had linear scleroderma with musculoskeletal involvement. Patients had been unsuccessfully treated with corticosteroids, methotrexate and/or mycophenolate mofetil. Abatacept was added to maintenance treatment with a disease-modifying drug; 13 patients were also receiving steroids at the start of abatacept treatment. After just 6 months, there was an improvement in the skin and a reduction in PGA-A. In most patients (80%), there was a significant improvement in skin and musculoskeletal activity at 12 months and every 6 months up to 24 months of follow-up. However, 16.7% of patients discontinued treatment due to adverse effects.

Abatacept proved to be an effective treatment, but the authors conclude that prospective studies are needed.

### 3.2. Antifibrinolytic Agents

The pathophysiology of morphea is based in part on abnormal signaling via the platelet-derived growth factor (PDGF) and TNFβ, which appears to be one of the key processes in the inflammatory response in morphea and systemic scleroderma [124].

A tyrosine kinase inhibitor (Imatinib) is approved for the treatment of chronic myelogenous leukemia and gastrointestinal stromal tumors. This drug interferes with both of these signaling pathways by blocking the activity of c-Kit, c-Abl and PDGF receptors [124].

In addition, Akmetshina et al. [125] concluded that imatinib may cause fibrosis regression by inhibiting collagen synthesis, leading to increased matrix degradation; hence its anti-fibrinolytic effect.

Coelho-Macias et al. [126] described the case of a 50-year-old patient with generalized scleroderma, diagnosed 10 years before the start of imatinib treatment; in addition, the patient developed multiple ulcers on his extremities during the course of the disease. For this reason, immunosuppressive treatment was not attempted, and the patient was only treated with pentoxifylline. The patient was started on imatinib for 12 months (3 months at 200 mg/day and 9 months at 300 mg/day). Clinical improvement was observed during treatment, with normal ulcer healing and reduction in inflammatory lesions, as confirmed by biopsy and ultrasound.

The limitations of the above trials is the small group size and the lack of a control group. These drugs are not without side effects, but no serious adverse events have been reported in the cases mentioned.

We have included all of the above medicines in Table 2.

## 4. Discussion

The key challenges for clinicians treating patients with morphea are early diagnosis and the earliest possible initiation of treatment to minimize future damage, such as cosmetic sequelae or limb deformity.

The identified immunological, genetic or epigenetic factors offer the opportunity to identify new therapeutic targets for the treatment of morphea subtypes that are resistant to standard therapies.

Small molecule biologics, used successfully in the treatment of rheumatological diseases, for example, could be an alternative therapy in morphea.

However, randomized, double-blind, placebo-controlled clinical trials are needed to provide answers about the efficacy and, above all, the safety of the new drugs and the possibility of treating the disease.

## Figures and Tables

**Figure 1 jcm-13-07134-f001:**
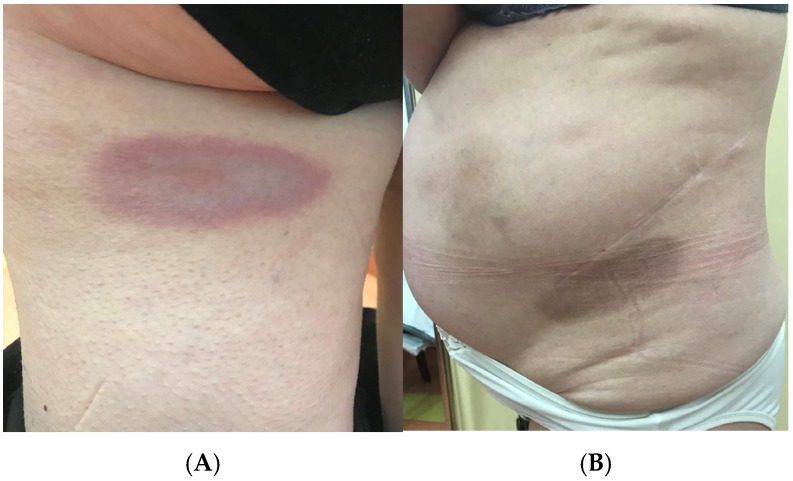
(**A**,**B**)—Clinical picture of morphea. (**A**)—active lesion, characteristic lilac ring, courtesy of Prof. Polańska.

**Figure 2 jcm-13-07134-f002:**
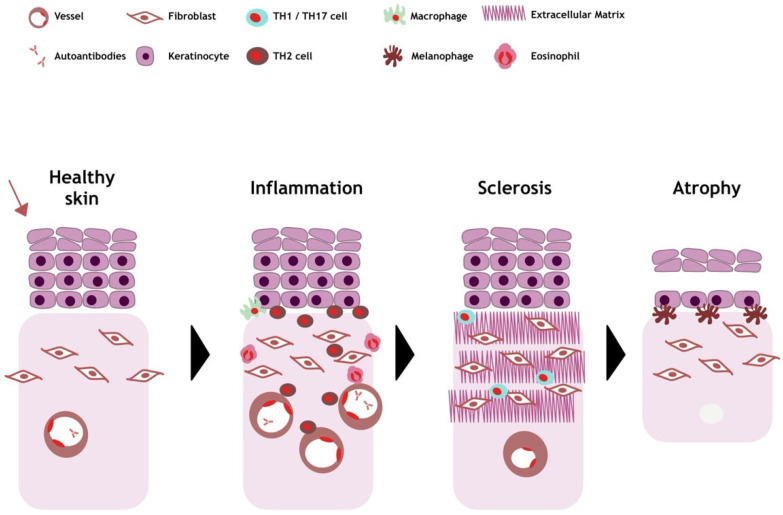
Pathogenesis of morphea. Factors such as infection and skin trauma (red arrow) can trigger inflammation in predisposed patients. T lymphocytes and eosinophils are involved in this process, appearing in the dermis and also in the vascular area. During the inflammatory phase, the endothelium is damaged, adhesion molecules appear and pro-inflammatory Th1 and Th17 cells are recruited, which activate fibroblasts via cytokines. Th2 cells then recruit T cells that produce profibrotic cytokines such as IL-6, IL-4. This leads to an increase in collagen fibres and subsequent sclerosis. The vessel walls thicken and the vessel lumen narrows. The final phase is atrophy, when the thickness of the epidermis is reduced and melanophages appear. There is a reduction in or atrophy of blood vessels and skin appendages. (The Figure 2 was made using Canva).

**Table 1 jcm-13-07134-t001:** Differential diagnosis of morphea [13,28,29].

**Localized (inflammatory lesions)** Lichen sclerosus Granuloma anulare Granulosarcoma Fixed erythema Urticaria pigmentosa Cutaneous radiation syndrome Chronic erythema migrans Mycosis fungoides
**Localized (hyperpigmentation lesions)** Cafe au lait spots Lichen planus induced by solar radiation Erythema dyschromicum perstans Post-inflammatory hyperpigmentation
**Localized (atrophic lesions)** Scarring Cellulitis Atrophic dermatitis of the extremities Lichen sclerosus
**Localized (sclerosis)** Circumscribed myxoedema Morpheaform basal cel carcinoma Necrobiosis lipoidica
**Linear** Glucocorticosteroids-induced lipoatrophy Site-specific lipoatrophy Progressive lipodystrophy
**Deep** Lupus panniculitis Inflammation of the subcutis
**Generalized** Systemic sclerosis GVHD Porphyria cutanea tarda Mixed connective tissue disease Scleredema adultorum of Buschke Scleromyxedema Nephrogenic systemic fibrosis

**Table 2 jcm-13-07134-t002:** New therapeutic options for morphea.

Authors	Number of Patients and Type of Morphea	Name of Drug	Prior Systemic/Topical Therapies	Dose of New Drug/or Median	Response and Total Time of Therapy/or Median	Adverse Events
Lythgoe (2018) [112]	5 P, age btw 6–13 y Linear morphea	Tocilizumab	MTX—5, MMF—5, ETP—2, CSA—1	8 mg/kg for *p* ≥ 30 kg 12 mg/kg for *p* < 30 kg 0, 2 and 4 wks, and then at 4-wk intervals	RT—after 6 mth TT—btw 12–25 mth	None
Lonowski et al. (2022) [115]	3 F, age btw 41–69 y Linear morphea—1, generalized morphea—2	Tocilizumab	PDN—2, MTX—2, MMF—1, MPDN—1, IVIG—1, RTX—1	162 mg/wk sc—2 8 mg/kg iv—1	Median response—3 months MT—44.6 mth	Hyperlipidemia—1
Zhang et al. (2019) [113]	1 F, 6 y Pansclerotic morphea	Tocilizumab	MTX, PDN, MPDN, MMF	300 mg every 4 wks	mLoSSI—22 to 6 PGA—30 to 17 TT—18 mth	None
Martini et al. (2017) [114]	1 F, 1 M (children) Pansclerotic morphea	Tocilizumab	MTX—2, PDN—2, MMF—2, MPDN—2, IMB—1	8 mg/kg every 4 wks	F—after 18 mth, LoSCAT from 58 to 47 TT—N/A	F—one episode of pneumonia
					M—TT—6 mth, after 30 mth LoSCAT from 57 to 43	
Li et al. (2018) [123]	18 P, mean age 13,4 y (12 F, 6 M) Linear morphea—12, Deep morphea—1, generalized morphea—1, mixed—5	Abatacept	MTX—17, MMF—16, GKS—18, HCQ—2, IMB—2, TA—2, LNLD—1, Biologics (ADA, INF, Tocilizumab)—1, PUVA—1	Median—10 mg/kg iv and sc	After 12 mth, 15 (83%) P MT—23.2 mth	Mood and/or behavioral issue—2
Fage et al. (2018) [122]	13 adults (11 F, 2 M) Morphea	Abatacept	N/A	500 mg < 60 kg or 750 mg > 60 kg iv on days 1, 15, 30 and thereafter every 4–6 wks	RT btw 3 to 32 mth Overall, an improvement/reduction in size of lesions or mRSS/mLoSSI score were observed Total—N/A	Fatigue 1–3 days after infusion, nausea, diarrhea (in some cases with fever) Withholding treatment—2 1-fatigue, tingling and aching sensation 2—colitis ulcerosa
Stausbøl-Grøn et al. (2011) [121]	47 y, F disseminated morphea	Abatacept	PCL—2, PDN—2, CSA—2, TA—1, UVA1—1	750 mg iv on days 1, 15 and 30, and thereafter every 4–6 wks (20 drug administration)	mRSS—18 to 1 TT-19 mth	1- Hypertension
	38 y, F Morphea profunda			500 mg iv on day 1, 15, 30, and thereafter every 4 wks	mRSS—13 to 6 TT—2.5 mth	treatment had to be stopped after 2.5 mth, patient had a breast cancer
Tang et al. (2023) [118]	6 y, F Linear morphea	Tofacitinib	MTX—1, TA—1, GLC—1, HCQ—1, UVA1—2, TCS—2	2 mth on 2.5 mg twice daily, 4 mth on 2.5 mg/d and 2 mth on 2.5 mg/d every other day	mLoSSI—6 to 1 LosDi—7 to 2 TT—8 mth	None
	13 y, M Linear morphea			2 mth on 5 mg twice daily, 3 mth on 5 mg/d	mLoSSI—7 to 1 LosDi—7 to 2 TT—5 mth	
Damsky et al. (2020) [119]	54 y, M Generalized morphea	Baricitinib	ECP, PDN	2 mg daily	Resolution of erythema after 2 mth Improvement in mobility after 6 mth TT—12 mth	None
Coehlo-Macias et al. (2014) [126]	50 y, M Generalized morphea with ulceration on the extremities	Imatinib	PTX	3 mth on 200 mg/d and 9 mth on 300 mg/d	Skin biopsy after 12—dermal thickness from 5 to 4 mm Improvement of joint mobility; left knee maximum extension went from initial 110° to 130° TT—12 mth	None

Abbreviations: P—Patient, y—Year old, MTX—Methotrexate, MMF—Mycophenolate-mofetil, ETP—Etanercept, wks—Weeks, wk—Week, RT—Response time, TT—Total time of therapy, mth—Month, Months, btw—Between, CSA—Cyclosporine, PDN—Prednisone, MPDN—Methylprednisolone, IVIG—Intravenous immunoglobulin, sc—Subcutaneous, iv—Intravenous, RTX—Rituximab, F—Female, M—Male, IMB—Imatinib, mLoSSI—The modified Localized Skin Severity Index, PGA—The Physician Global Assessment, LoSCAT—The Localized Scleroderma Assessment Tool, GKS—Glucocorticoids, HCQ—Hydroxychloroquine, TA—Topical agents, LNLD—Lenalidomide, ADA—Adalimumab, INF, Infliximab, PUVA—Psoralen plus ultraviolet A photochemotherapy, mRSS—The modified Rodnan skin score, PCL—Penicillamine, GLC—Glycyrrhizin, TCS—Tacrolimus ointment, LosDi—Localized scleroderma damage index, MT—Median time of therapy, ECP—Extracorporeal photopheresis, PTX—Pentoxifylline, N/A—not available.

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
