# Peer review of "What Is New in Morphea—Narrative Review on Molecular Aspects and New Targeted Therapies"

_jcm, 2024, doi:10.3390/jcm13237134_

Round 1
Reviewer 1 Report
Comments and Suggestions for Authors
A very nice review . The title nr. 3 Treatment maybe it would be better on the same page as the content of the chapter. Table 1 title should be on the same page as the table. The columns of the table should permit that the letters of the names should de on the same line
Author Response
Dear Reviewer,
Thank you for giving us the opportunity to submit a revised draft of the manuscript “What is new in morphea – narrative review on molecular aspects and new targeted therapies" for publication in the Journal of Clinical Medicine. We appreciate the time and effort that you dedicated to providing feedback on our manuscript and are grateful for the insightful comments on and valuable improvements to our paper.
We have improved the layout of the manuscript and suggestions for the table.

Reviewer 2 Report
Comments and Suggestions for Authors
In this manuscript the Authors provided an interesting review about Morphea, an autoimmune connective tissue disease characterized by excessive collagen deposition in the dermis and subcutaneous tissue. The manuscript is well written and organized. However, I have some points to be addressed:
1) The Authors should add an image about etiopathogenesis of Morphea, which reassumes the main process involved
2) The Authors should add a brief paragraph about differences between Morphea and Systemic sclerosis, since many of the process involved in the pathogenesis of the latter are present also in Morphea
Author Response
Dear Reviewer,
thank you very much for your comments on our manuscript. Please see the attachment.

Reviewer 3 Report
Comments and Suggestions for Authors
Dear Authors!
Thank you for the opportunity to review your manuscript.
Morphea is the most common presentation of localized scleroderma. Authors provide the narrative review including different aspects of the pathogenesis and treatment.
According the manuscript I have several suggestions
1) You used term linear morphea. What did you mean? Linear scleroderma or envoupe de sabre? You manuscript is focused on the morphea, but sometime you included data about pansclerotic morhea, linear scleroderma etc. I think you might focus only in morphea and exclude the other forms of the localized scleroderma or change the manuscript title to localized scleroderma and expanded to other data and forms. Now in your manuscript the data about linear form is very scarse, but these patients required different management, e.g. brain MRI ef linear scleroderma affected the face and neck.
2) Your manuscript is focused on adults, but sometime you included the data about the kinds, but the principles are differents. PUVA treat,ent is not a first choice in pedoiatric morphea, compared to adults. Please provide age-specific manage,ent or concise in adult form only
3) Please add the role of thermography and ultrasound in the diagnostics
4) The list of the differential diagnosis is required, especially in terms of morphological skeen involvement (deep etc). Please provide the figure comparing different tissue depth involvement in different forms of the scleroderma-like diseases.
5) Please provide the data about the lichen
6) In the treatment section, please add information about the rituximab treatment, especially if you left the other forms of the localized scleroderma. e.g. the principles of the management of the forms with brain involvement (encoup de sabre forms)
7) Please provide the indication for the systemic treatment
8) Please provide the data about cosmetic surgery, indications. time?
9) Please provide the data about fat or stemm cell injections as the possible treatment of the morphea
10) The diagnostic and treatment algorithms (two separate figures) will be very nice
Author Response

(The authors gave the same response as above.)

Round 2
Reviewer 2 Report
Comments and Suggestions for Authors
The Manuscript is improved enough and it is now suitable for publication
Reviewer 3 Report
Comments and Suggestions for Authors
Dear Authors!
Thank you for your response and revised version of the manuscript.
I have no additional queries.